# An Assessment of Women’s Knowledge of the Menstrual Cycle and the Influence of Diet and Adherence to Dietary Patterns on the Alleviation or Exacerbation of Menstrual Distress

**DOI:** 10.3390/nu16010069

**Published:** 2023-12-25

**Authors:** Anna Ciołek, Malgorzata Kostecka, Julianna Kostecka, Paulina Kawecka, Monika Popik-Samborska

**Affiliations:** 1Department of Chemistry, Faculty of Food Science and Biotechnology, University of Life Sciences, Akademicka 15, 20-950 Lublin, Poland; anna.ciolek@up.lublin.pl (A.C.); paulina.kawecka@up.lublin.pl (P.K.); 2Faculty of Medicine, Medical University of Lublin, Chodźki 19, 20-093 Lublin, Poland; kostecka.julianna@gmail.com; 3Dietetic Centre Project Health, Ametystowa 16, 20-576 Lublin, Poland

**Keywords:** menstrual cycle, menstrual disorders, dysmenorrhea, visual analog scale (VAS), menstrual health, nutrition knowledge, anti-inflammatory diet, proinflammatory diet

## Abstract

A growing number of women of reproductive age experience menstrual disorders. The menstrual cycle is considerably influenced by dietary habits, physical activity, and the use of stimulants. The main aim of this study was to assess women’s knowledge about the menstrual cycle and the influence of diet and lifestyle factors on menstrual symptoms, and to identify dietary models that may alleviate or exacerbate menstrual distress. A total of 505 young women participated in the study. Nearly 90% of the respondents reported at least one menstrual disorder, mostly dysmenorrhea (70.7%), whereas secondary amenorrhea was least frequently reported (13.8%) (*p* = 0.002). In the study population, dysmenorrhea/menstrual distress was linked with higher consumption frequency of certain food groups. Women with severe dysmenorrhea consumed refined cereal products, processed meat, sugar, and water significantly more frequently than women with moderate menstrual pain. In turn, sweetened dairy products, animal fats, and fruit were consumed more frequently by women with low intensity of menstrual pain (mild pain). Significant differences in knowledge about the menstrual cycle and physiological changes in the body were observed between the compared dietary models.

## 1. Introduction

A growing number of women of reproductive age experience menstrual disorders. Menstrual symptoms such as dysmenorrhea, heavy menstrual bleeding, and perimenstrual mood disorders are known to be widespread among the general population [1]. Clinical trials have shown that menstrual symptoms can have a large impact on women’s mental and physical well-being, personal relationships, education and career attainment, and can lead to long-standing impairments in the quality of life [2]. Furthermore, it is estimated that many women initially do not consult a doctor while facing menstrual symptoms [3,4]. However, dysmenorrhea, which is characterized by menstrual pain in the absence of pelvic pathology, is one of the main reasons for gynecological consultations among young women [5]. The annual economic burden due to menstrual symptoms in Japanese women was estimated at approximately 683 billion Japanese yen, and in more than 70% of cases, dysmenorrhea was associated with a decrease in work productivity [6,7]. Data indicate that more than one-quarter of women reduced their working hours or were absent from work for at least 1 day per 6 months as a result of menstrual pain [6]. Depending on their severity, menstrual distress symptoms can also negatively affect women’s daily lives by preventing them from participating in social activities and maintaining social relations [7,8,9]. Research has also demonstrated that work or school absenteeism due to menstruation poses a significant economic, social, and health burden [9,10,11,12]. Despite the widespread prevalence of menstrual distress and the resulting decrease in the quality of life, including school and work absenteeism, this illness has received remarkably little scientific attention.

Dietary and lifestyle factors significantly affect the functioning of body organs and sex hormone levels. The menstrual cycle is considerably influenced by dietary habits, physical activity, and the use of stimulants. These factors affect the BMI and body fat percentage which are directly linked with ovulation disorders, and painful and irregular menstruation. The importance of dietary patterns for pain relief in women with dysmenorrhea is increasingly discussed in the literature [13]. However, too little attention has been paid to the role of specific food components in reducing menstrual distress. It is believed that healthy eating patterns may have pain-relieving properties. The development of an optimal diet for patients with dysmenorrhea constitutes a major challenge. Diets with low nutrient density, low in vitamins and minerals, and abundant in processed foods, may exert a negative impact on women’s overall health and well-being, and increase the risk of diet-related diseases which, under certain circumstances, can lead to menstrual disorders and exacerbate menstrual distress symptoms [14,15].

In recent years, numerous attempts have been made to identify modifiable factors, including lifestyle factors, that potentially alleviate menstrual distress. Research findings could substantially contribute to an improvement in women’s well-being by analyzing the relationship between dietary components and lifestyle factors that affect menstrual distress symptoms. The main aim of this study was to assess selected parameters associated with menstrual health and to correlate these parameters with the studied women’s diet and nutrition knowledge. Potential correlations between adherence to an emergent dietary model and the severity of menstrual distress were also examined.

## 2. Materials and Methods

### 2.1. Study Design

A cross-sectional study was conducted in the city of Lublin and in Lublin voivodeship in eastern Poland between April 2022 and January 2023. The study involved a paper questionnaire and an online questionnaire that was completed independently by the participants with the use of Google Forms.

Participants

The minimal sample size for the study was calculated with the use of Cochran’s formula [16] for the Polish population of women aged 18 to 40 years (N = 5,144,208 as of 31 December 2022) based on Statistical Yearbook data (Statistics Poland) [17]. The minimal sample size was determined at 384 persons at a significance level of 0.05, maximum error of 5%, and a sampling fraction of 0.5. The calculated sample size was adjusted for a 20% attrition rate to produce a total of 461 participants. A higher number of female respondents was achieved during recruitment, which decreased the maximum error and increased the power of the applied tests.

Inclusion criteria:-Age: 18–40 years;-Absence of serious illnesses that constitute exclusion criteria;-Informed consent to participate in the study.

Exclusion criteria:
-Amenorrhea (absence of menstruation);-Miscarriage in the previous 6 months;-Pregnancy;-Presence of biological offspring;-Use of hormonal drugs (other than contraceptives), steroids, antidepressants, sleeping pills, or sedatives in the previous 6 months or directly before the trial;-Chronic diseases and metabolic disorders, including type 1 and 2 diabetes, insulin resistance, polycystic ovary syndrome, endometriosis, sterility, infertility, hormonal disorders, thyroid disorders, kidney disorders, asthma, Crohn’s disease, anemia, eating disorders, epilepsy, rheumatoid arthritis, cancer, HPV infection;-Diagnosed gynecological diseases or disorders, including congenital abnormalities of the reproductive tract;-Pelvic floor disorders;-Lack of consent to participate in the study.

### 2.2. Study Tools and Data Collection

Information about the study was posted on Facebook groups, posters at Lublin universities, and in gynecological clinics. The survey was completely voluntary and anonymous, and the participants did not provide any personal or sensitive information that could enable their identification. The study involved a conventional paper questionnaire and an online questionnaire that was completed with the use of Google Forms on a dedicated Facebook page. All participants were residents of Lublin voivodeship. Before the trial, all women were informed about the purpose of the study and the research protocol. Survey completion was regarded as an indication of consent to participate in the study. The main questionnaire contained questions about the menstrual cycle (duration; presence, type and prevalence of menstrual symptoms; heavy bleeding; pain; intermenstrual bleeding; secondary amenorrhea), as well as 7 questions analyzing the participants’ knowledge about the physiology of the menstrual cycle and menstrual disorders. The participants received 1 point for every correct answer and 0 points for every incorrect answer. There were a maximum of 7 points in total, where 0–4 points denoted a low level of knowledge and >4 points denoted a high level of knowledge. Menstrual pain was rated on the visual analog scale (VAS). The participants were asked to indicate the average severity of menstrual pain by placing an “x” on a straight horizontal line with a length of 10 cm. The ends of the line represented the extreme limits of pain, where 0 points denoted a complete absence of pain, and 10 denoted severe pain. Pain intensity was self-assessed on the following scale: low (0–4.0 points), moderate (4.1–7.0 points), and severe (7.1–10 points), where 0.1 points was represented by 0.1 cm on the horizontal line. This part of the questionnaire also contained questions about the use of vitamin D supplements, dietary supplements (including herbal supplements), smoking, and alcohol consumption. The second part of the study involved a short food frequency questionnaire (FFQ). The questionnaire contained 37 food products divided into the following groups: cereal products; dairy products; meat and meat products; eggs and fish; fats and bread spreads; vegetables; pulses; fruit; nuts; sugar and confectionery products; beverages (including alcoholic beverages); and fast-food products. The respondents were asked to indicate the average consumption frequency of the products listed in the FFQ in the past year by selecting one of the seven options in the questionnaire: less than once a month; 1–3 times a month; 1–2 times a week; 3–4 times a week; 5–6 times a week; once daily; twice or more daily. The declared consumption frequency was converted into daily frequency using the appropriate conversion factors (Table 1). The participants’ biographic data were collected, including age, place of residence, education, and marital status. The surveyed subjects were also asked to indicate their body height (cm) and body weight (kg) based on independently performed anthropometric measurements and to describe any changes in body weight in the previous 6 months. Based on the obtained anthropometric data, the body mass index (BMI) was calculated by dividing body weight (kilograms) by the square of the height (meters). The interpretation of BMI was based on the guidelines of the World Health Organization:-<18.5 kg/m^2^—underweight;-18.5–24.99 kg/m^2^—normal weight;-25.0–29.99 kg/m^2^—overweight;-30.0–34.99 kg/m^2^—stage I obesity;->35 kg/m^2^—stage II obesity.

Paper questionnaires were filled out independently at home. The questionnaires were revised, and missing information was provided during a meeting with a researcher. Online questionnaires were completed with the use of Google Forms, and the completeness of the answers was checked by the researchers. The paper questionnaire and the online questionnaire involved the same main questionnaire and the FFQ. A total of 571 women completed the questionnaire, and 505 correctly completed questionnaires were returned.

### 2.3. Dietary Patterns

Dietary patterns were derived by cluster analysis. The input variables were 14 dietary (in times/day) components of DPs. All input variables were standardized using our own database to achieve a mean equal to 0 and a standard deviation equal to 1. To identify the optimal number of clusters, the analysis was conducted several times. The K-means clustering algorithm was used, and the subjects were grouped based on the Euclidean distance. Finally, two clusters were selected (an anti-inflammatory model and a proinflammatory model). The correctness of cluster identification and labelling was verified by comparing the components of DPs between clusters in a one-way analysis of variance.

### 2.4. Data Processing and Statistical Analysis

Missing data were addressed by excluding incomplete questionnaires or sections from the analysis. The results were presented in tables and figures and were described. Categorical variables were presented as sample percentages (%), and continuous variables were expressed as the mean ± standard deviation. The differences between groups were analyzed in the chi-squared test (categorical variables) or the Mann–Whitney test (continuous variables). The Kruskal–Wallis test was applied to analyze the relationships between the variables in more than two mutually independent groups. Before statistical analysis, the data were checked for normal distribution in the Kolmogorov–Smirnov test. The odds ratios (ORs) and 95% confidence intervals (95% CIs) were calculated. The reference category (OR = 1.00) was mild intensity of menstrual pain. The significance of ORs was assessed by Wald’s statistics. The results of all tests were regarded as statistically significant at *p* < 0.05. Data were processed in the Statistica program (version 13.1 PL; StatSoft Inc., Tulsa, OK, USA; StatSoft, Krakow, Poland) and in SPSS v. 27.

## 3. Results

The study involved 505 women with the mean age of 24.7 ± 4.5 years (range: 19–40 years). Nearly 50% of the participants resided in cities with a population of ≥250,000, and rural inhabitants accounted for 27.9% of the studied group (Table 2). Most of the surveyed women lived in informal relationships (49.9%), and a relatively high percentage of the population (23.0%) declared to be single. The vast majority of the studied subjects were professionally active or were students in paid employment (84.2%), and nearly 5% of the participants were unemployed.

The average BMI was 22.4 ± 4.2 kg/m^2^. Most of the studied subjects (69.2%) had a healthy body weight. Overweight and obesity were determined in 23.5% of the surveyed population, where 51.3% of the participants were characterized by gynoid obesity (pear-shaped body), and 48.7% by android obesity (apple-shaped body). Only 5.2% of the subjects were underweight. Changes in body weight in the previous 6 months were reported by 52.0% of the participants, where 70% of the subjects declared weight gain, and 30% reported weight loss (*p* = 0.013). In both cases, the average change in body weight was 3.8 ± 1.6 kg. Changes in body weight were more frequently reported by women with excessive BMI (*p* = 0.001) and women with android obesity (*p* = 0.021).

### 3.1. Selected Menstrual Health Parameters

In the vast majority of the surveyed subjects, the length of the menstrual cycle ranged from 26 to 31 days, and cycles shorter than 26 days were reported by 10.7% of the participants. Most women had regular menstrual cycles (84.3%), and menstruation lasted 5.1 ± 1.0 days on average. Bleeding intensity was assessed as moderate by the vast majority of the participants (63.2%), whereas heavy or very heavy bleeding was reported by nearly every fourth respondent (26.1%). The average pain intensity measured on the VAS was 6.3 ± 2.8 points, and the results were used to divide women into three groups with different levels of pain intensity. The first group consisted of women with a VAS score of 0–4 points (mild pain), the second group comprised participants with a VAS score of 4.1–7 points (moderate pain), and the third group consisted of subjects with a VAS score of 7.1–10 points (severe pain).

Menstrual distress and disorders significantly affect overall health status. Nearly 90% of the respondents reported at least one menstrual disorder, mostly dysmenorrhea (70.7%), whereas secondary amenorrhea was least frequently reported (13.8%) (*p* = 0.002). The most frequent menstrual distress symptoms were abdominal pain, back pain, and generalized pain, and the prevalence of these symptoms increased with a rise in the menstrual pain score (Table 3).

### 3.2. The Influence of Diet and Eating Habits on the Menstrual Cycle

Most of the subjects consumed four meals per day (N = 207, 41%), whereas the smallest number of women (N = 29, 5.7%) consumed two or fewer meals per day. In the studied group, 16.0% of the respondents adhered to a special diet, mostly a vegetarian diet (54.5% of the respondents adhered to a special diet). Light (low-calorie) food products were consumed by 73.3% of the surveyed subjects, and every tenth respondent consumed these products regularly or always. Dietary supplements were used by 84% of the surveyed population in the preceding 6 months. The most popular supplements were vitamin D, magnesium, vitamin B6, vitamin and mineral preparations, omega-3 fatty acids, and herbal supplements that alleviate menstrual pain and boost immunity. The phase of the menstrual cycle considerably affected food intake in more than half of the respondents (*p* = 0.012). During menstruation, 21.7% of the women (*p* = 0.023) experienced increased hunger, whereas 17.6% (*p* = 0.031) experienced food cravings, mainly sugar cravings.

On average, most women consumed 1730 mL of beverages per day, around 3.4 vegetable portions per day, 2.5 portions of cereal products per day, and 1.6 portions of sweets per day. The most frequently consumed food products were beverages, vegetables, fruit, sugar, and sweets, whereas fish, dry legumes, and nuts were consumed least frequently (Table 4). It should be noted that the consumption frequency of these groups of food products varied considerably from several times per day to several times per month. The following groups of food products were consumed least frequently: fermented dairy products, alcohol, fish, potatoes, and dry legumes. In this context, only reduced alcohol consumption can be regarded as a positive factor.

In the study population, dysmenorrhea/menstrual distress was linked with a higher consumption frequency of certain food groups. Women with severe dysmenorrhea consumed refined cereal products, processed meat, sugar, and animal fats significantly more frequently than women with mild menstrual pain. In turn, sweetened dairy products, whole-grain cereal products, fruit, and vegetables were consumed more frequently by women with mild intensity of menstrual pain (mild pain) (Table 4).

### 3.3. Evaluation of Menstrual Health Parameters and Eating Habits in Women Adhering to Two Dietary Models

Two dietary models were developed based on the groups of food products and the frequency with which these products were consumed by the surveyed women. The first model, referred to as the anti-inflammatory model, is largely consistent with the planetary health diet which is composed mainly of vegetables, fruit, whole-grain cereal products, and legumes, and is characterized by reduced consumption of red meat, processed meat, refined cereal products, starchy vegetables, and added sugar. The second model, referred to as the proinflammatory model, involved higher consumption of processed foods, in particular refined cereal products, processed meat, animal fats, potatoes, sugar, sweets, sweetened beverages, and fast foods, and reduced consumption of vegetables, fruit, dairy products, and high-fiber cereal products. Based on the declared consumption of various groups of food products, 324 women (64.6%) were included in the first model, and 181 women (35.4%) were included in the second model (*p* = 0.0231). Smoking and alcohol consumption were the main health-related parameters that differentiated the participants. The number of women who had never smoked and consumed alcohol sporadically (less than several times per month) was significantly higher in model 1 than in model 2 (65.6% vs. 41.9%, *p* = 0.027, and 52.1% vs. 34.8%, *p* = 0.031, respectively).

No significant differences in average BMI values were observed between women following model 1 and model 2 (22.7 ± 4.6 vs. 22.3 ± 3.8 kg/m^2^, respectively, *p* = 0.236). However, the number of women with a healthy BMI was significantly higher in model 1 than in model 2 (79.1% vs. 63.0%, *p* = 0.004). The percentage of overweight and obese women was significantly higher in model 2 than in model 1 (37.0% vs. 15.9%), but no significant differences in the percentage of women with gynoid and android obesity were noted between models.

The percentage of women experiencing low intensity of menstrual pain (mild pain) was significantly higher in model 1 (41.9%) than in model 2 (6%) (Table 5). Women adhering to the anti-inflammatory diet were also less affected by other menstrual distress symptoms.

The use of dietary supplements, consumption of food products with a decreased content of one or more nutrients, and consumption of low-calorie foods did not differ significantly between the compared models (*p* > 0.05). Women adhering to the proinflammatory diet experienced hunger significantly more often (*p* = 0.28) and more frequently consumed a higher number of meals per day (*p* = 0.038) during menstruation. Women from the compared models differed significantly in their knowledge about the menstrual cycle and physiological changes in the body (*p* < 0.05). Women adhering to the anti-inflammatory diet had higher levels of knowledge.

Education was a factor that significantly influenced the participants’ knowledge about the effect of dietary components such as saturated fatty acids (Figure 1), vitamin D (Figure 2), and simple sugars (Figure 3) on the menstrual cycle.

Most participants were of the opinion that a diet rich in unsaturated fatty acids (54.1%, *p* = 0.0941), vitamin D (68.9%, *p* = 0.0231), zinc, selenium, and iodine (68.9%, *p* = 0.162), B vitamins (76.4%, *p* = 0.0712), and antioxidants (66.7%, *p* = 0.172) contributes to a healthy menstrual cycle. Only 36.4% of the surveyed women identified saturated fatty acids and 35.4% identified simple sugars as dietary components that do not contribute to a healthy menstrual cycle.

## 4. Discussion

The literature on the relationship between diet and menstrual distress is selective and ambiguous. According to research, there is evidence to suggest that diet is linked with menstrual distress, including dysmenorrhea and/or other menstrual distress symptoms. However, most of the published studies are of low quality and rely on different methodologies (for diagnosing menstrual pain or collecting information about the participants’ diets), which prevents a reliable comparison of data and the formulation of conclusions. Most studies on the subject were conducted in Asia and Africa, whereas only a handful of trials were carried out in Europe, including in Spain [18], Serbia [19], and Italy [20,21], which limits the scope of the available data. Two Polish epidemiological studies demonstrated that dysmenorrhea affects 65% of Polish women [22,23]. In the work of Barcikowska et al. [24], only 68 of the studied subjects (6.0%) did not report any symptoms of dysmenorrhea. In contrast, Szymańska et al. [13] assessed the influence of adherence to a dietary model on menstruation distress only in women with heavy and painful menstruation. The present study contributes new knowledge about the influence of diet and the proposed dietary models on the menstrual cycle and distress in a population of Polish women.

Recent studies have reported that in women who do not receive hormonal therapy (contraception, hormone replacement therapy), hormonal changes during the menstrual cycle have a significant impact on eating habits. An awareness of these changes can help women control their food intake and maintain normal weight [25,26]. The premenstrual syndrome (PMS) may predispose women to changes in appetite and food cravings [26,27]. In the population analyzed in this study, nearly a quarter of the surveyed women experienced increased appetite, including sugar cravings, during menstruation, and high pain intensity was significantly correlated with increased consumption of sweets (*p* = 0.001). Similar results were reported by Lefebvre et al. [28] who analyzed women’s preferences for various groups of food products during the menstrual cycle. They found that most menstruating women craved fruit, sweets, and carbohydrates [28]. One of the hypotheses to be considered for the increase in carbohydrate intake during menstruation is the relationship between simple carbohydrates (high glycemic index) and a higher production of cerebral serotonin which reduces the negative mood effects [29]. It is believed that women unconsciously increase carbohydrate consumption in the days prior to menstruation to produce neurotransmitters that are related to mood improvement because carbohydrates, especially simple carbohydrates, increase the availability of tryptophan, a precursor of serotonin in the brain [29,30,31].

According to many researchers, dietary factors, lifestyle factors, and dietary models that potentially affect the intensity of menstrual pain should be identified. Research has shown that lifestyle habits, including the frequency of breakfast consumption, were associated with menstrual pain. In the present study, women with mild menstrual pain consumed regular meals (74.1% vs. 69.3%, *p* = 0.0031) and breakfasts (63.9% vs. 58.1%, *p* = 0.046) more frequently than women with severe menstrual pain. Similar results were noted in a study of Japanese women aged 18–20 years, where respondents who consumed breakfast 0–3 times per week experienced irregular menstrual cycles and severe menstrual pain significantly more often than women who consumed breakfast 4–7 days per week [32,33].

Menstrual health is integral to women’s overall health and well-being. Despite the above, menstrual disorders remain insufficiently investigated, and some women have very little knowledge about the menstrual cycle. In the Western world, women tend to be more knowledgeable, and menstruation is no longer regarded as a taboo subject [34], but the physiological basis of menstruation, biological changes at puberty, the menstrual cycle, and infection risks posed by poor menstruation hygiene are hardly ever discussed openly [35,36]. A study conducted in Bangladesh revealed that most adolescent girls lacked scientific knowledge about menstruation and puberty, and adolescents were reluctant to address this topic or seek treatment for menstrual distress due to popular myths, misconceptions, and other cultural biases [37]. In a study of 125 American women, 52.8% of the participants had a high knowledge about the ovulatory cycle, whereas only 50.4% of the respondents were familiar with the average duration of menstrual flow [38]. Lungdsberg [39] investigated 1000 American women of various ethnicity, socioeconomic status, and educational background, and found that approximately 40% of the respondents were not fully familiar with the ovulatory cycle. A survey of 14- to 21-year-old girls and women conducted by Plan International UK in the United Kingdom and Ireland revealed that one in seven participants did not know what was happening during their first menstruation [40]. In 2020, a study carried out in Poland by the Kulczyk Foundation also demonstrated that nearly 40% of the surveyed girls and women aged 15–45 years had insufficient knowledge about the menstrual cycle, limited access to reliable information about menstruation, and a decreased quality of life and well-being [41]. In the present study, more than 60% of the participants had sufficient knowledge about the physiology of the menstrual cycle (education was a significantly differentiating factor, *p* < 0.002; 73% of university graduates vs. only 36% of women with secondary school education had adequate levels of knowledge), but more than a third of the respondents were not familiar with the length of the menstrual cycle, and 24% could not indicate the successive phases of the cycle.

Many women relied on alternative methods of self-care that were not always optimal. Similarly to this study, Fernández-Martínez [42] observed that non-steroidal anti-inflammatory drugs were the main self-care strategy of alleviating menstrual pain. In an epidemiological study of Danish women, the prevalence of menstrual pain (dysmenorrhea) was found to be inversely associated with the dietary intake of fish oil (*n*-3 fatty acid) and vitamin B12. These findings support the hypothesis that menstrual cramps, which are prostaglandin-mediated, can be influenced by dietary fatty acids, and that fish oil supplements could be potentially used to treat or prevent dysmenorrhea [43]. A high ratio of dietary omega-3 to omega-6 fatty acids could potentially alleviate dysmenorrhea. According to Hansen [44], diets rich in polyunsaturated fatty acids (PUFAs) can decrease the consumption of non-steroidal anti-inflammatory drugs during menstruation. In the current study, more than 80% of the surveyed women used dietary supplements, including remedies that alleviate menstrual pain and discomfort.

Adherence to a dietary pattern could be a factor that affects health, including menstrual health. However, this issue has been investigated by a small number of reliable studies [13,45,46]. Similarly to the present study, Szymanska et al. [13] found that adherence to an unprocessed diet based on the Mediterranean diet or the anti-inflammatory diet reduced pain intensity or decreased the incidence of menstrual distress. The cited authors analyzed the consumption frequency of selected food groups, such as vegetables, fruit, dairy, meat, legumes, fish, fast foods/salty products, and sweets. The presented findings can guide the development of new therapies for relieving menstrual pain.

### Strengths and Limitations

Despite the fact that menstrual distress, in particular dysmenorrhea, affects a large portion of the female population, only a few studies have addressed the relationship between menstrual distress symptoms and diet. Most of the studies have been conducted in Asia on small population samples, and they investigated only a limited number of food products. The present study has several strengths. Above all, attempts were made to examine the complex relationship between menstrual distress symptoms, in particular dysmenorrhea, and Polish women’s diets. In addition, the study sample was selected based on a narrow set of inclusion criteria, and validated tools were used assess menstrual pain and the participants’ diets. The results can be used to draw reliable conclusions about the influence of diet on menstrual distress and to explore the underlying mechanisms of action. Two dietary models were identified and correlated with menstrual health factors, lifestyle factors, and women’s knowledge about the menstrual cycle. The present findings can be used to formulate the objectives of health and nutrition education programs for women of reproductive age. Furthermore, moderate and severe menstrual pain was reported by a relatively high number of the surveyed women, which increased the power of statistical tests.

The main limitation of the study is that the study sample was not randomly selected, and that cross-sectional data were collected. As a result, the causes of menstrual distress could not be identified, and only the presence of correlations or relationships between the analyzed factors in the studied group of women could be determined.

## 5. Conclusions

Can menstrual health be linked to women’s diet and nutritional knowledge? Can adherence to a dietary pattern confer potential benefits and influence the severity of menstrual discomfort? This study showed that in the analyzed population, menstrual pain/distress was associated with the consumption frequency of selected groups of food products. Two dietary models that potentially affect the menstrual cycle were identified. Consumption of a diet based on the principles of the planetary diet promoted a reduction in menstrual discomfort. Women consuming an anti-inflammatory diet containing low-processed products (vegetables, fruit, whole-grain cereals, dairy and legumes) experienced significantly less menstrual pain. In contrast, consumption of highly processed foods, including sweets, processed meat, foods high in saturated fat and low in fiber, was more frequently associated with increased menstrual pain and discomfort.

Significant differences in knowledge about the menstrual cycle and physiological changes in the body were observed between women adhering to the compared dietary models. The results indicate that an educational program targeting women of reproductive age and health professionals should be implemented to disseminate knowledge on the impact of diet and lifestyle factors on the alleviation of menstrual distress symptoms.

## Figures and Tables

**Figure 1 nutrients-16-00069-f001:**
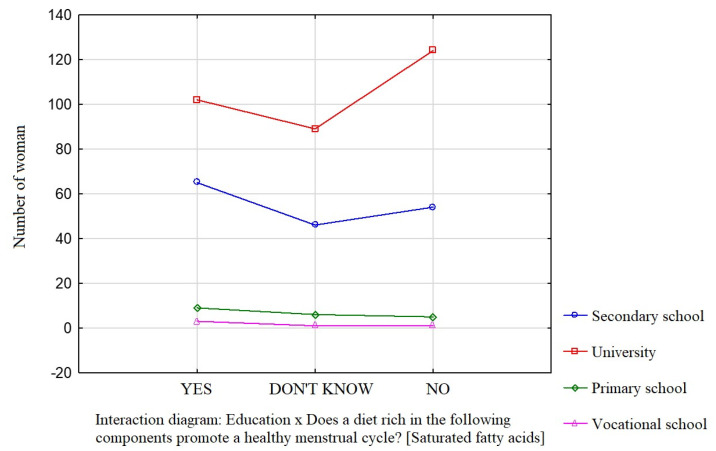
The relationship between education and the participants’ knowledge about the influence of dietary components on the menstrual cycle (saturated fatty acids).

**Figure 2 nutrients-16-00069-f002:**
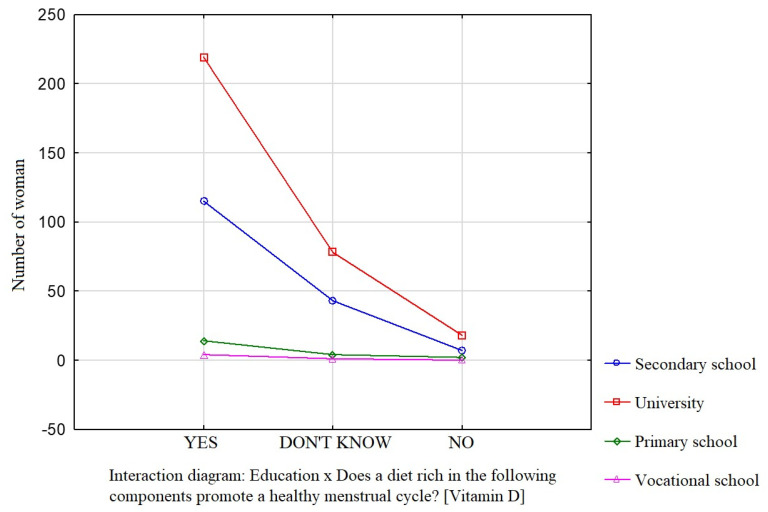
The relationship between education and the participants’ knowledge about the influence of dietary components on the menstrual cycle (vitamin D).

**Figure 3 nutrients-16-00069-f003:**
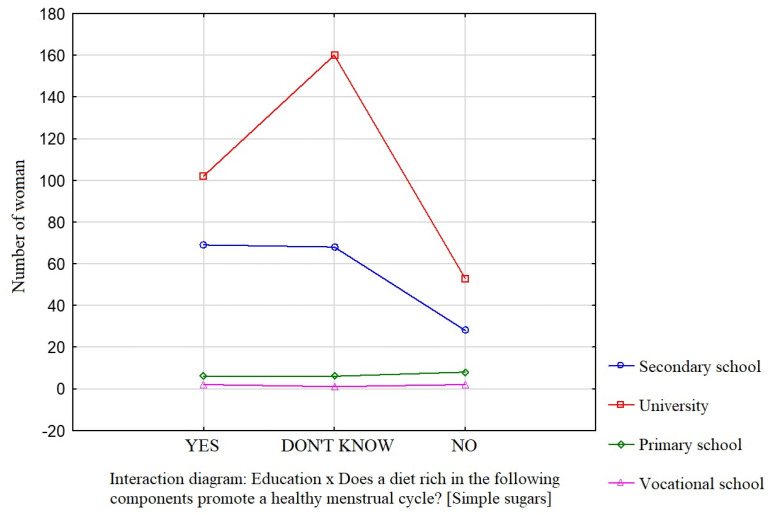
The relationship between education and the participants’ knowledge about the influence of dietary components on the menstrual cycle (simple sugars).

**Table 1 nutrients-16-00069-t001:** Conversion factors applied to the consumption frequency classes defined in the FFQ.

Frequency of Consumption as Defined in FFQ	Conversion Factor to Daily Frequency
Less than once a month	0.00
1–3 times a month	0.07
1–2 times a week	0.21
3–4 times a week	0.50
5–6 times a week	0.79
Once daily	1.00
Twice or more daily	2.00

**Table 2 nutrients-16-00069-t002:** Sociodemographic characteristics of the studied group (N = 505).

Sociodemographic Parameters	N	N [%]	*p*-Value
Place of residence			<0.05
City with a population of >250,000	254	50.3%	
City with a population of <250,000	110	21.8%	
Rural area	141	27.9%	
Age			<0.05
<25 years	198	39.2%	
>25 years	307	60.8%	
Marital status			<0.05
Single	116	23.0%	
Informal relationship	252	49.9%	
Married	137	27.1%	
Education			<0.05
Primary school	20	4%	
Secondary school	165	32.7%	
University	315	62.3%	
Vocational school	5	1%	
Professional status			<0.05
Student	57	11.3%	
Student in paid employment	159	31.5%	
Employed	266	52.7%	
Unemployed	23	4.5%	

Differences are significant at *p* < 0.05.

**Table 3 nutrients-16-00069-t003:** Menstrual disorders and distress symptoms relative to the self-reported intensity of menstrual pain (VAS).

	Mild Pain, N = 147 (29.1%)		Moderate Pain, N = 234 (46.4%)		Severe Pain, N = 124 (24.5%)		Kruskal–Wallis Test
	N	N%	N	N%	N	N%	
Abdominal pain	89	60.5	177	75.6	124	100	0.003
Anxiety	19	12.9	159	67.9	75	60.5	0.005
Generalized pain	23	15.6	173	73.9	117	94.3	0.002
Uterine contractions	106	72.1	170	72.6	95	76.6	ns
Back pain	91	61.2	178	76.0	97	78.2	ns
Headache	56	38.1	56	23.9	68	54.8	0.012
Fatigue	39	26.5	79	33.7	84	67.7	0.003
Leg pain	27	18.4	141	60.3	96	77.4	0.002
Nausea	11	7.5	17	7.3	10	8.0	ns
Insomnia	29	19.7	59	25.2	87	70.2	0.003
Loss of appetite	46	31.3	72	33.2	41	33.1	ns
Diarrhea/constipation	51	34.7	71	30.3	43	34.6	ns
Heart palpitations	12	8.2	24	10.3	27	21.8	0.041

ns—not statistically significant.

**Table 4 nutrients-16-00069-t004:** Consumption frequency of selected groups of food products relative to the intensity of menstrual pain (tertile division, N = 505) and the odds ratios (95% confidence interval) in an analysis of the relationships between severe menstrual pain vs. mild menstrual pain and the consumption of particular food groups.

			Intensity of Menstrual Pain						*p*-Value	
	Total N = 505		Mild		Moderate		Severe			Severe Menstrual Pain (Ref. Mild Menstrual Pain)
	Daily Frequency		Daily Frequency		Daily Frequency		Daily Frequency			
Refined cereal products	1.4	(0–2)	1.2	(0–2)	1.5	(0–2)	1.7	(0–2)	0.0031	1.76 ** (1.06–1.98)
Whole-grain cereal products	0.81	(0–2)	0.94	(0–2)	0.83	(0–2)	0.67	(0–2)	0.023	0.88 * (0.64–1.02)
Milk	1.5	(0–2)	1.5	(0–2)	1.4	(0–2)	1.4	(0–2)	ns	0.92 (0.89–1.14)
Fermented milks	0.3	(0–2)	0.3	(0–2)	0.4	(0–2)	0.3	(0–2)	ns	0.94 (0.91–1.09)
Sweetened dairy products	1.5	(0–2)	1.5	(0–2)	1.7	(0–2)	1.0	(0–2)	0.0016	0.73 (0.62–1.07)
Fresh meat	0.6	(0–2)	0.6	(0–2)	0.7	(0–2)	0.6	(0–2)	ns	1.02 (0.97–1.04)
Processed meat	1.5	(0–2)	1.3	(0–2)	1.3	(0–2)	1.7	(0–2)	0.002	1.54 * (1.12–1.87)
Fish	0.2	(0–1)	0.2	(0–1)	0.2	(0–1)	0.2	(0–1)	ns	1.03 (0.92–1.09)
Vegetable fats	0.4	(0–2)	0.4	(0–2)	0.5	(0–2)	0.3	(0–2)	ns	0.91 (0.81–1.01)
Animal fats	1.6	(0–2)	1.4	(0–2)	1.3	(0–2)	1.8	(0–2)	0.041	1.45 * (1.12–1.87)
Potatoes	0.3	(0–0.79)	0.3	(0–0.79)	0.3	(0–0.79)	0.3	(0–0.79)	ns	1.02 (0.97–1.04)
Vegetables	1.7	(0–2)	1.8	(0–2)	1.5	(0–2)	1.1	(0–2)	0.0027	0.53 ** (0.46–0.89)
Dry legumes	0.3	(0–1)	0.3	(0–1)	0.3	(0–1)	0.3	(0–1)	ns	1.02 (0.97–1.05)
Nuts	0.2	(0–2)	0.2	(0–2)	0.2	(0–2)	0.2	(0–2)	ns	1.02 (0.98–1.07)
Fruit	1.7	(0–2)	1.9	(0–2)	1.6	(0–2)	1.4	(0–2)	0.0027	0.71 * (0.64–0.89)
Sugar and sweets	1.6	(0–2)	1.1	(0–2)	1.4	(0–2)	1.9	(0–2)	0.001	2,19 ** (1.34–2.51)
Sweetened beverages	0.5	(0–2)	0.3	(0–2)	0.7	(0–2)	0.4	(0–2)	0.0031	1.19 (1.01–1.36)
Fast foods	0.7	(0–1)	0.7	(0–1)	0.8	(0–1)	0.7	(0–1)	ns	1.02 (0.97–1.08)
Alcohol	0.4	(0–2)	0.6	(0–2)	0.4	(0–2)	0.5	(0–2)	0.042	0.91 (0.76–1.09)
Water	1.7	(0–2)	1.6	(0–2)	1.1	(0–2)	1.8	(0–2)	0.0026	1.23 (1.01–1.46)

Statistically significant: ns—not statistically significant; * *p* < 0.05; ** *p* < 0.01.

**Table 5 nutrients-16-00069-t005:** Menstrual health parameters, diet, and levels of knowledge in the compared dietary models.

	Model 1 (Anti-Inflammatory)		Model 2 (Proinflammatory)		*p*-Value in the Chi-Squared Test
	N = 324	N%	N = 181	N%	
Intensity of menstrual pain					0.015
mild	136	41.9%	11	6%	
moderate	167	51.5%	67	37%	
severe	21	6.6%	103	57%	
Regular menstrual cycle					0.041
yes	289	89.2%	137	75.7%	
no	35	10.8%	44	24.3%	
Menstrual distress symptoms					0.017
<3	111	34.3	35	19.3%	
3–8	186	57.4%	67	37%	
>8	27	8.3%	79	43.6%	
Number of meals per day					0.038
1–3	89	27.5%	46	25.5%	
4	151	46.6%	56	30.9%	
>5	84	25.9%	79	43.6%	
Adherence to a special diet					0.028
yes	33	10.2%	48	26.5%	
no	291	89.8%	133	73.5%	
Consumption of light food products					0.225
never	81	25%	54	29.8%	
rarely	212	65.4%	120	66.3%	
regularly/always	31	9.6%	7	3.9%	
Use of dietary supplements in the previous 6 months					0.317
yes	263	81.2%	161	89%	
no	61	18.8%	25	11%	
Increased food intake during menstruation					0.037
yes	112	34.6%	106	58.6%	
no	212	65.4%	75	41.4%	
Increased hunger during menstruation					0.028
yes	79	24.4%	94	51.9%	
no	245	75.6%	87	48.1%	
Knowledge about dietary components that influence mood during menstruation					0.021
yes	247	76.2%	102	56.4%	
no	77	23.8%	79	43.6%	
Knowledge about the menstrual cycle					0.036
low	105	32.4%	98	54.1%	
high	219	67.6%	83	45.9%	
The influence of diet on the men-strual cycle					0.029
no influence	0	-	13	7.2%	
low influence	54	16.7%	36	19.9%	
considerable influence	124	38.3%	75	41.4%	
very high influence	146	45%	57	31.5%	

## Data Availability

Due to ethical restrictions and participant confidentiality, data cannot be made publicly available.

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
