# Peer review of "An Assessment of Women’s Knowledge of the Menstrual Cycle and the Influence of Diet and Adherence to Dietary Patterns on the Alleviation or Exacerbation of Menstrual Distress"

_nutrients, 2023, doi:10.3390/nu16010069_

Round 1
Reviewer 1 Report
Comments and Suggestions for Authors
Dear authors,
Your manuscript is well written. The impact of diet and lifestyle on menstrual symptoms is very useful to all reproductive women.
On the meantime, at the introduction I would like to see data not only from worldwide but from Europe and especially your country Poland.
I also have not noticed any Ethics Committee Permission to confirm the study was under the Declaration of Helsinki.
Conclusions are well structured but somehow they could be expanded to suggest specific food products to be consumed for relieving menstrual symptoms and distress.
Author Response
Dear Reviewer,
Thank you for reviewing our article. Your feedback and insights have been immensely helpful in improving the quality of the manuscript. We appreciate the time and effort dedicated to reviewing our work. We look forward to incorporating your suggestions to make the paper even stronger. Our detailed responses are presented below. All changes have been marked in blue and red font in the manuscript.
|
On the meantime, at the introduction I would like to see data not only from worldwide but from Europe and especially your country Poland. |
Polish studies were presented in the Discussion section. However, the available literature is limited because most studies conducted on the Polish population focused on the perimenopausal period or women with endometriosis who were excluded from the present study. |
|
I also have not noticed any Ethics Committee Permission to confirm the study was under the Declaration of Helsinki. |
The study was approved by the Ethics Committee of the Medical University of Lublin. The relevant information was provided in the revised manuscript. |
|
Conclusions are well structured but somehow they could be expanded to suggest specific food products to be consumed for relieving menstrual symptoms and distress. |
The Conclusions section was expanded, and the planetary diet and a low-processed diet were referenced as dietary models for relieving symptoms of menstrual distress. |

Reviewer 2 Report
Comments and Suggestions for Authors
Ciołek et al. They present an interesting and relevant manuscript in the field of research. It is a coherent study with adequate justification. However, it is necessary to address major points:
-The authors use a pretentious title. The authors must focus it on the specific context of this Journal.
-The authors must improve the summary, with information on the results and their translation.
-Please, authors must increase the number of keywords.
-The authors must make an introduction to the state of the art in a unitary manner and with a current bibliography. The authors must justify the importance of these studies.
-The authors must specify the calculation of the sample size with statistical methods.
-The authors must include the statistical potential.
-The authors must justify the statistical method and test. Does it follow a normal distribution? The steps to use the statistical test are not clear.
-The authors must severely improve the presentation of the results. This point is absolutely unacceptable in the current format.
-The figures are very difficult to see. This reviewer has to use the magnifying glass to see figure 1, 2 and 3.
-The data must be better described.
-The authors must mention and justify the protocol with international guidelines.
-The discussion is very poor. The authors should improve it with translational data and focus it in the context of women's health.
-A graphic summary is necessary.
-The authors must greatly improve the use of English grammar with experts.
Comments on the Quality of English LanguageEnglish very difficult to understand/incomprehensible.
Author Response
Dear Reviewer,
Thank you for all constructive remarks. The Reviewer's comments have enabled us to improve the quality of the manuscript. We hope that the revised manuscript will provide new insights into the under-researched relationship between dietary factors and menstrual distress symptoms. All changes have been marked in blue and red font in the manuscript.
|
-The authors use a pretentious title. The authors must focus it on the specific context of this Journal. |
The title of the manuscript was modified. |
|
-The authors must improve the summary, with information on the results and their translation. |
The proposals were rewritten and expanded. |
|
Please, authors must increase the number of keywords. |
The number of keywords was increased. |
|
-The authors must make an introduction to the state of the art in a unitary manner and with a current bibliography. The authors must justify the importance of these studies. |
The introduction was restructured and additional references were provided. |
|
-The authors must specify the calculation of the sample size with statistical methods. |
The minimum sample size was calculated according to known and recommended formulas, and it was checked in Statistica. |
|
-The authors must include the statistical potential. -The authors must justify the statistical method and test. Does it follow a normal distribution? The steps to use the statistical test are not clear. |
The description of statistical analyses was improved and expanded. In our previous studies, we relied on a similar approach to describe the results of statistical analyses. |
|
-The authors must severely improve the presentation of the results. This point is absolutely unacceptable in the current format. The data must be better described. |
We agree that the results presented in Table 3 (Table 4 in the revised manuscript) were difficult to understand and interpret. To improve the presentation of results, an additional table (Table 1) was provided, the results in Table 4 were recalculated, and the odds ratios (95% confidence interval) in the analysis of the relationships between severe menstrual pain vs mild menstrual pain and the consumption of particular food groups were presented in Table 4. In our previous work, the same approach was used to present the results of the FFQ and the odds ratios (95% confidence interval). |
|
-The figures are very difficult to see. This reviewer has to use the magnifying glass to see figure 1, 2 and 3. |
Drawings were enlarged. |
|
-The authors must mention and justify the protocol with international guidelines. |
Thank you for this comment. However, to the best of our knowledge, there are no protocols or international guidelines for describing the effect of dietary patterns on menstrual distress. The procedure of analyzing the results of the FFQ and assessing anthropometric parameters was described in greater detail. |
|
-The discussion is very poor. The authors should improve it with translational data and focus it in the context of women's health. |
Thank you for this comment. In the Discussion section, we have added extended information on conducting similar research in Poland. We showed that there are few of them and the potential benefits for women's health and improving their quality of life are very large. Based on a review of the literature, dietary ingredients that can alleviate symptoms of menstrual distress and promote women's health were described in the Discussion. A similar approach to structuring the discussion of results was used in our previous studies. |
|
-A graphic summary is necessary. |
A graphical abstract was provided. |
|
-The authors must greatly improve the use of English grammar with experts. English very difficult to understand/incomprehensible. |
Since the authors are not native speakers of English, the manuscript has been translated by a professional translator who has extensive experience in editing scientific manuscripts. It has also been spell-checked and grammar-checked by a native English speaker prior to re-submission. To facilitate the peer review process, the manuscript has been revised to increase the clarity of presentation. Grammar, spelling and style have been checked. Should you have any further remarks concerning the use of English in the paper, we would appreciate it if you could provide us with specific examples of errors that in your opinion should be corrected. |

Round 2
Reviewer 2 Report
Comments and Suggestions for Authors
Minor editing of English language required.
Comments on the Quality of English LanguageMinor editing of English language required.
Author Response
We hope that all the corrections made have improved the quality of our manuscript. As suggested by the Reviewer, we have carried out another linguistic revision. The manuscript was checked again by a professional translator and a native English speaker prior to resubmission.